
**Debris flow modeling at Meretschibach and Bondasca catchment,**
**Switzerland: sensitivity testing of field data-based erosion model**
Florian Frank[1], Brian W. McArdell[1], Nicole Oggier[2], Patrick Baer[3], Marc Christen[4] and Andreas
Vieli[3]
[1] Swiss Federal Institute for Forest, Snow and Landscape Research, Birmensdorf, 8903,
Switzerland
[2] wasser/schnee/lawinen, Ingenieurbüro André Burkard AG, Brig-Glis, 3900, Switzerland
[3] Glaciology, Geomorphodynamics & Geochronology, Department of Geography, University of
Zurich, Zurich, 8057, Switzerland
[4] WSL Institute for Snow and Avalanche Research SLF, Davos Dorf, 7260, Switzerland
*Correspondence to*: Florian Frank (florian.frank@wsl.ch)
**Abstract**
Debris flow volumes can increase due to the incorporation of sediment into the flow as a
consequence of channel-bed erosion along the flow path. This study describes a sensitivity analysis
of the recently-introduced RAMMS debris flow entrainment algorithm which is intended to help
solve problems related to predicting the runout of debris flows. The entrainment algorithm predicts
the depth and rate of erosion as a function of basal shear stress based on an analysis of erosion
measurements at the Illgraben catchment, Switzerland (Frank et al., 2015). Starting with a
landslide-type initiation in the RAMMS model, the volume of entrained sediment was calculated
for recent well-documented debris-flow events at the Bondasca and the Meretschibach catchments,
Switzerland.  The sensitivity to the initial landslide volume was investigated by systematically
varying the initial landslide volume and comparing the resulting debris-flow volume with estimates
from the field sites. In both cases, the friction coefficients in the RAMMS runout model were
calibrated using the model where the entrainment module was inactivated. The results indicate that
the entrainment model predicts plausible erosion volumes in comparison with field data. By
including bulking due to entrainment in runout models, more realistic runout patterns are predicted
in comparison to starting the model with the entire debris-flow volume (initial landslide plus
entrained sediment). In particular, lateral bank overflow – not observed during this event – is
prevented when using the sediment entrainment model, even in very steep ($\approx$ 60–65 %) and narrow
(4–6 m) torrent channels. Predicted sediment entrainment volumes are sensitive to the initial
landslide volume, suggesting that the model may be useful for both reconstruction of historical
events as well as the modeling of scenarios as part of a hazard analysis.



## 1. Introduction

Sediment erosion caused by debris flows strongly influences the bulking behavior of debris-flows (Iverson, 1997). The entrainment of sediment along the channel has been observed to considerably increase the volume of debris flows at many different locations (e.g. Hungr et al., 2005; Scheuner et al., 2009; Iverson et al., 2010; Berger et al., 2010a; Berger et al., 2011; Schürch et al., 2011; Iverson et al., 2011, McCoy et al., 2012; Tobler et al., 2014; Frank et al., 2015). Two recent extreme examples from the central Swiss Alps in the last decade showed significant bulking along the flow path. In the Spreitgraben catchment (2009-2011), the overall multi-surge event volumes increased to about 90'000 to 130'000 m$^3$ – mainly due to erosion along the active channel on the fan (Tobler et al., 2014; Frank et al., 2015). At the Rotlauigraben catchment (2005), about 2/3 of the total volume of 550'000 m$^3$ was eroded from the debris-flow fan during a multiple-surge debris-flow event initiated by the failure of a glacier moraine during an intense rainfall event (Scheuner et al., 2009). Therefore, the debris-flow erosion and bulking process should be included in debris-flow runout models to increase the accuracy of runout predictions including the overall runout distance, location and amplitude of lateral bank overflow but also – importantly for hazard assessment – the flow and depositional pattern on the fan (Gamma, 2000; Scheuner et al., 2009; Hussin et al, 2012; Han et al., 2015; Frank et al., 2015).

However, models which include bulking by debris flows are relatively new and their performance for practical applications has not yet been systematically investigated. Most entrainment modeling studies focused on the field site where the erosion data for the underlying entrainment modeling concept was collected and/or exclusively dealt with a single model application field site to test their concept for entrainment modeling (e.g. Han et al., 2015; Frank et al., 2015). Herein we describe the systematic application of the new RAMMS entrainment/bulking model (Frank et al., 2015) for several recent events in the Swiss Alps.

Computational debris-flow runout models, which usually neglect erosion, are often used to assess runout distance and pattern (Crosta et al., 2003; D'Ambrosio et al., 2003; Medina et al., 2008; Hungr and McDougall, 2009; Christen et al., 2012) and are therefore useful for hazard analysis where predictions of flow intensity (e.g. the spatial distribution of flow depth and velocity) are required (e.g. Scheuner et al., 2011). Because the debris flow process often was observed to cause significant entrainment of sediment which can strongly influence the flow (e.g. Dietrich and Dunne, 1978; Suwa and Okuda, 1980; Gallino and Pierson, 1984; Hungr et al., 1984; Benda, 1990; Pierson et al., 1990; Meyer and Wells, 1997; Vallance and Scott, 1997; Berti et al., 1999; Cannon and Reneau, 2000; Fannin and Wise, 2001; May, 2002; Wang et al., 2003; Revellino et al., 2004; Scott et al., 2005; Godt and Coe, 2007; Breien et al., 2008; Gartner et al., 2008; Guthrie et al., 2010; Procter et al., 2010; Berger et al., 2010; Berger et al., 2011; Schürch et al., 2011; Iverson et al., 2011; McCoy et al., 2012; Tobler et al., 2014; Frank et al., 2015), the importance of including entrainment and bulking debris flow runout modeling would be appropriate. Processed-based





entrainment rates using algorithms which consider the material properties of the debris flow bulk
(Crosta et al., 2003; D'Ambrosio et al., 2003; Medina et al., 2008; Deubelbeiss and McArdell,
2012) as well as pre-specified entrainment rates which pre-define the absolute volume of eroded
material (Beguería et al., 2009; Hungr and McDougall, 2009; Hussin et al., 2012) have been
introduced in numerical runout models.
Recently, we introduced an erosion algorithm in the RAMMS debris flow runout model for the
assessment of debris flow erosion and bulking (Frank et al., 2015). The erosion algorithm uses a
relation between basal shear stress and erosion based on an analysis of data from the Illgraben
catchment, Switzerland (Frank et al., 2015; Berger et al., 2011; Schürch et al., 2011). The
entrainment model was used to predict the overall erosion pattern and erosion volume at the first
site where it was tested, the Spreitgraben, Switzerland. However, secondary erosion processes such
as bank collapse and small torrential flood events between the debris flow events increased the
uncertainty in the evaluation of the model. As a consequence, additional sensitivity tests were not
made. In this study we therefore focus on testing the sensitivity of the RAMMS debris flow and
entrainment model by assessing the sensitivity of total event volume (initial landslide volume plus
volume of eroded sediment) to initial flow volume. This is especially important in hazard analysis
where landslide scenarios are considered to trigger debris flows. For this sensitivity analysis, we
evaluated two Alpine catchments with diverse topography and recent well-documented debris
flows with volumes up to a few 10,000 m³: the Bondasca catchment in Southeastern Switzerland
and the Meretschibach catchment in Southern Switzerland.
**2.   Erosion modeling study sites and available data**
**2.1. Meretschibach catchment, Switzerland**
The Meretschibach catchment is located in Southern Switzerland, adjacent to and east of the
Illgraben catchment (Figure 1). The catchment area is about 9.2 km² and ranges from the summit of
the Bella Tola mountain (3,025 m a.s.l.) to the confluence with a drainage channel (619 m a.s.l.)
following into the Rhone River. Debris flows in the Meretschibach currently originate mainly in
the Bochtür subcatchment (1.42 km² area) which is covered mostly by steep debris slopes with
hillslope angles on the talus deposits of up to 60%. Patches of forest are present below the treeline
(2,200 m a.s.l.) and at the margins of the catchment, and largely contiguous forest is found along
both sides of the channel below an elevation of 1,600 m. The Bochtür subcatchment is underlain by
Triassic sericitized quartzite and white quartzites of the Bruneggjoch formation (Gabus et al. 2008).
The surface has several terrace-like structures have been mapped as sacking-type features (Gabus
et al., 2008) and are likely sources of landslides and rockfall.
Sediment deposits are abundant on the steep slopes of the catchment, originating from a variety of
mass wasting processes. Field observations of rockfall, the presence of damaged trees, and



unpublished records in the community forestry archives records indicate that rockfall is a dominant
process for generating sediment. Observations in the source area also indicate that dry ravel of
gravel and sand is also common in the summer months when the hillslopes are relatively dry.
According to the event inventory debris flows occur mainly between April and October (Szymczak
et al. 2010). Small debris flows start and deposit in the upper catchment, often depositing at an area
of lower slope located an elevation of approximately 2,000 m a.s.l. Convective storms or long
duration rainfall events have been observed to mobilize these sediment deposits and initiate debris
flows.
Georadar profiles on the west side of the unforested part of the Bochtür subcatchment as well a
airborne georadar measurements indicate that the sediment deposits are up to 5 m thick
(Fankhauser et al., 2015), although independent observations of the spatial distribution of sediment
thickness are not available. However extrapolation of that value to other parts of the catchment
must be made with caution because the profiles were made on a talus deposit, which may be
interpreted as a depositional area on the hillslope, that exhibits little geomorphic evidence of
debris-flow activity.
In the years 2013 and 2014 several instruments and devices were installed in the catchment. In
October 2013, a meteorological station was installed above the initiation zone to measure
precipitation, temperature and snow height. Inexpensive wildlife-observation cameras recorded
images every 15 minutes during daylight were positioned along the most active western channel to
document the changes along the active channel. A debris flow monitoring station was installed on
23 July 2014 (Oggier et al. 2015a). It consisted of three geophones and a radar to measure the flow
stage. The radar is triggered by the geophones or the meteorological station and provides detailed
recordings of the debris flow hydrograph at a resolution of 1 Hz.
During summer 2014, three debris flows occurred. Because the monitoring station was installed
after the first event (20 July 2014), no hydrograph data are available for this event. Precipitation
and hydrograph data for the debris flow events on 28 and 29 July 2014 indicate that the debris flow
event on 28 July was triggered due to convective storms with large rainfall intensity (up to 3.3 mm
/ 10 min) while the event 29 July 2014 initiated after a few hours of steady rainfall with moderate
intensity (up to 1.5 mm / 10 min). The pictures from camera 4 (see Fig. 1 for the location) clearly
showed that the initiation of the event on July 28 took place between 19:45 and 20:15 (UTC +2),
corresponding with the hydrograph measured at the observation station.
To obtain additional information about the initial volume and the spatial distribution of erosion, the
height models from 15 July and 28 October were compared. The digital elevation model of 17 July
was the result of a photogrammetry flight by swisstopo. The second digital elevation model (28
October) – which is a surface model (including vegetation) – was taken with a drone (Oggier et al.
2015b). The results indicate that the volume of the events eroded at the open debris slopes of



Bochtür was between 800 and 1,200 m$^3$. Due to additional erosion downslope of the Bochtür
subcatchment, the total volume of the debris flow events was between 8,000 and 10,000 m$^3$.
**2.2. Bondasca catchment, Switzerland**
The Bondasca catchment in south-eastern Switzerland is a tributary to the Bergell valley (Figure 2).
The catchment area covers about 20.9 km$^2$ . The geology is dominated by the Tertiary intrusion of
the Bergell granite. Originating from within the North wall of Pizzo Cengalo, a rock avalanche on
27 December 2011 deposited about 1.5 10$^6$ m$^3$ of sediments in the upper catchment with a runout
of up to two kilometers from the rock wall. The deposits are up to 17 m thick and cover an area of
about 0.760 km$^2$ while the hydrological sub-catchment is about 1.18 km$^2$ defined by the point
where the channel leaves the rock avalanche deposits at the lower end of the deposit.
The sudden sediment input from the rock avalanche was followed by several debris flows in the
summer of 2012 (5 and 14 July, 25 August, 24 September) whereof the two events in July
evacuated about 90'000 m$^3$ of sediments from the rock avalanche deposit. The debris flows
originated mainly just below a flat-shaped rock face. Some of the debris flow surges are thought to
have been triggered due to water accumulation at the toe of the wall causing firehose-type debris
flow initiation (Figure 3B and 5B) e.g. as described by Godt and Coe (2007). The slope of the
channel on the rock avalanche deposit varies between approx. 32° (≈ 71 %) below the flat-shaped
rock face and regularly decreases to 15° (≈ 33 %) at the lower end of the rock avalanche deposit.
**3.  Debris-flow entrainment modeling**
The goal of this study is to evaluate the erosion algorithm implemented in the RAMMS debris flow
model (version 1.6.25) which has been previously described by Frank et al. (2015). In particular,
the sensitivity of the predicted erosion to the input parameters will be investigated, and the data sets
described above provide a new basis for evaluating the model. The previous study (Frank et al.,
2015) focused on demonstrating that more realistic runout results can be achieved when including
sediment entrainment and bulking into the runout model. However that study also left many
unanswered questions regarding the sensitivity of the model to input parameters, especially the
initial landslide volume, which was not possible to assess in the previous study. Herein we focus on
describing the sensitivity of the model to the initial landslide volume, using the two well-
documented events described earlier in the paper.
Although the RAMMS model and the erosion algorithm have been published elsewhere, they will
be briefly described below to provide the necessary background information for understanding the
model. The underlying numerical formulas of shallow water equation and the Voellmy friction
approach used in the RAMMS debris flow model are presented in detail in Christen et al. (2010);
the field-data based empirical entrainment model is described in Frank et al. (2015).





### 3.1. Computational debris-flow model RAMMS

The RAMMS debris-flow model is based on 2D depth-averaged shallow water equations for
granular flows in three dimensions given by the coordinates of the topographic surface of the
digital elevation model in a Cartesian coordinate system $(x, y, z)$ and at time $(t)$ (Bartelt et al.,
1999; Christen et al., 2010). The mass balance equation incorporates the field variables flow height
$H (x, y, t)$ and flow velocity $U (x, y, t)$ and is given by
$$\dot{Q} (x, y, t) = \partial_t H + \partial_x(HU_x) + \partial_y(HU_y) .\tag{1}$$
where $\dot{Q} (x, y, t)$ describes the mass production source term and $U_x$ and $U_y$ represent the depth-
averaged velocities in horizontal directions x and y (Christen et al., 2010). The depth-averaged
momentum balance equations account for the conservation of momentum in two directions x and y:
$$S_{g_x} - S_{f_x} = \partial_t(HU_x) + \partial_x \left( c_x HU^2{}_x + g_z k_{a/p} \frac{H^2}{2} \right) + \partial_y\left(HU_xU_y\right) ,\tag{2}$$
$$S_{g_y} - S_{f_y} = \partial_t\left(HU_y\right) + \partial_x\left(HU_xU_y\right) + \partial_y \left( c_y HU^2{}_y + g_z k_{a/p} \frac{H^2}{2} \right) .\tag{3}$$
where the earth pressure coefficient $k_{a/p}$ is normally set to 1 when running the standard Voellmy-
Salm friction approach, $c_x$ and $c_y$ represent topographical coefficients determined from the digital
elevation model, $S_g$ is the effective gravitational acceleration, and $S_f$ the frictional deceleration in
directions x and y (Christen et al., 2010). The frictional deceleration $S_f$ of the flow is determined
using the Voellmy friction relation (Salm et al., 1990, and Salm, 1993) and specifies the Coulomb
friction µ scaling with the normal stress and the turbulent friction ξ depending on the velocity
squared (Christen et al., 2012; Bartelt et al., 2013):
$$S_f = \mu \cdot \rho \cdot Hg\cos(\phi) + \frac{\rho g U^2}{\xi}\tag{4}$$
where ρ is the mass density, g is the gravitational acceleration, ϕ is the slope angle (approximately
similar to the internal friction angle of the material), and $Hg\cos(\phi)$ is the normal stress on the
overflowed surface. The tangent of the effective internal friction angle of the flow material can be
defined for the resistance of the solid phase (the term containing µ) which extensivly controls
deceleration behavior of a slower moving flow. On the other hand, the resistance of the viscous or
turbulent fluid phase (the term including ξ) prevails for a quicker moving flow (Bartelt et al.,

201  2013).





### 3.2. Debris-flow entrainment model


The entrainment model was constructed using field data from the Illgraben catchment in
Switzerland (Frank et al., 2015). The entrainment model describes the maximum erosion depth as a
function of channel-bed shear stress and the vertical erosion rate of channel-bed sediment erosion.
In detail, the model is based on the analysis of differential elevation models from pre- and post-
event DTMs by Schürch et al. (2011b). This provides the depth of net erosion in a cell as a function
of the local shear stress acting on the channel bed at the base of the flow. Similarly, the rate of
erosion is constrained to be at the rate reported by Berger et al., 2011, using *in situ* erosion sensors,
also at the Illgraben channel. In the analysis of Schürch et al (2011b), flow heights were determined
using values interpolated between lateral levees after each event and the shear stress τ is
approximated using the depth-slope product:
$$\tau = \rho g h S \tag{5}$$
where ρ is the bulk mass density of the flow, h is flow height, and S is the channel slope. An
approximation of the typical potential erosion depth at the Illgraben follows the 50% percentile line
fit to the distribution of elevation change for four debris flow events (Fig. 3a in Schürch et al.,
2011b). The erosion algorithm implemented in the RAMMS entrainment model is defined by the
maximum potential erosion depth $e_m$ and a specific erosion rate. The relationship between the shear
stress estimated and the measured erosion (Schürch et al., 2011b) is described as a linear function
of shear stress using a proportionality factor $\frac{dz}{d\tau}$ (Eq. 2). The maximum potential erosion depth $e_m$ is
calculated using a critical shear stress $\tau_c$ (= 1 kPa) and the proportionality factor $\frac{dz}{d\tau}$ (= 0.1 m kPa$^{-1}$)
as a function of basal shear stress τ:
$$e_m = \begin{cases} 0 \ for \ \tau < \tau_c \\ \frac{dz}{d\tau}(\tau - \tau_c) \ for \ \tau \geq \tau_c \end{cases} \tag{6}$$
The average rate of erosion recorded at the erosion sensor site during the Illgraben debris flow
event of 1 July 2008 (Berger et al., 2011) is used to define a specific erosion rate $\frac{dz}{dt}$.
$$\frac{dz}{dt} = -0.025 \ for \ e_t \leq e_m \tag{7}$$
When the critical shear stress $\tau_c$ is exceeded, sediment can be entrained from the channel.
Entrainment stops when the actual erosion depth $e_t$ reaches the maximum potential erosion depth
$e_m$ (Eq. 2). Normally, the specific erosion rate is implemented using the default value $\frac{dz}{dt} =$
$-0.025 \ \text{ms}^{-1}$ (Eq. 3) as presented in Frank et al. (2015). However, the model also allows to



account for larger or smaller erosion scenarios by either doubling the rate or cutting it in half. In
this study, we will use these variable erosion rates for testing the sensitivity of the model.

### 3.3. Erosion model setup

#### 3.3.1. Topographic resolution

This study focuses on the evaluation of the sensitivity of the predicted (modeled) channel-bed
erosion in relation to the initial volume (e.g. initial landslide size) and the comparison of the model
results and the erosion pattern observed in the field. The ability to reproduce the observed erosion
patterns highly depends on a realistic representation of the channel morphology where the channel
is clearly visible in the DTM (Deubelbeiss et al., 2010 and 2011; Scheuner et al., 2011; Hohermuth
and Graf, 2014) and the channel dimensions (e.g. cross-sectional area) in the DTM have to be
similar to what is observed in the field (e.g. Frank et al., 2015). In this study, the initial topographic
data available for the Meretschibach catchment (described above) are on a square grid of 0.5 m for
a channel with a width of 2 to 4 m. At the Bondasca catchment data are available on a 2 m square
grid for channel varying in width from about 5 to 20 m. Although a channel width to DTM grid
spacing ratio of more than 5 to 10 would probably produce more accurate results, such data are
generally unavailable and the increase in the time for a simulation would be impractical.

#### 3.3.2. Erosion model starting condition: block release and input hydrograph

The type of initial release mechanism, lock release or input hydrograph, can be determined based
on field observations, potential model constraints and previous modeling experience using the
RAMMS debris flow model (Bartelt et al., 2013). Recent debris flow modeling studies
(Deubelbeiss et al., 2010; Deubelbeiss et al., 2011; Han et al., 2015) summarized that debris flows
in steep channels are mostly triggered by the sudden destabilization of material originating from
lateral bank collapses or dam-type deposits located within the channel itself. Han et al. (2015)
concluded that a hypothetical scenario such as the breaking of a dam – which they used to start
their erosion model simulations – provides a stable and consistent release method. Deubelbeiss et
al. (2010 and 2011), for a case study in the Swiss Alps, suggested that the block release method is
most appropriate method for small to moderate initial volumes ranging from 1 m$^3$ up to 100 m$^3$
using the RAMMS debris flow model. The alternative release method using a discharge
hydrograph seems to be more suitable for larger initial volumes (Deubelbeiss et al., 2010 and 2011)
(> 100 m$^3$) which – in general – might be plausible for the larger channel of the Bondasca
catchment.
The main problem with the block release is that the initial flow depth, width, or length of the initial
landslide can be unrealistically large in comparison to field observations. Users have to resort to
such large initial landslide volumes because most models do not allow for erosion along the





channel path. The total debris flow volume, typically measured in the deposition zone, is often used
as the initial landslide volume, thereby implicitly ignoring the possibility that channel-bed erosion
and flow bulking occur (Frank et al., 2015). The input hydrograph starting condition in RAMMS
was intended to help circumvent this problem by allowing users to specify an influx of debris as a
function of time at a point lower in the watershed (e.g. just above the fan apex).
The block release volume is calculated by defining a specific block release height (with a precision
of 1 cm) based on a pre-defined release area. The model assumes an instantaneous failure of the
landslide. The initial landslide surface elevation is then set to the initial elevation of the land
surface using an automatic procedure in RAMMS (the "subtract release from DTM" option in
RAMMS introduced in version 1.6.45). The main advantage of this procedure is that it prevents
unrealistic lateral spreading of the initial landslide mass in comparison with a landslide "block"
situated on top of the land surface.

### 3.3.3. Specified erosion rates

As a basis for comparison of the sensitivity of the erosion algorithm, we hold constant the default
erosion model coefficients (critical shear stress $\tau_c$, potential erosion depth as a function of basal
shear stress $\frac{dz}{d\tau}$, erosion rate $\frac{dz}{dt}$) described above. In the previous study (Frank et al., 2015) we
demonstrated that an erosion rate of $\frac{dz}{dt}$ = 2.5 cm s$^{-1}$ based on field data from the Illgraben
catchment, Switzerland (Berger et al., 2011) produces plausible results for the much steeper
Spreitgraben catchment. The catchments described in this paper are different in size and slope, so
one might expect some variation in erosion rate. However, the erosion algorithm in RAMMS
allows for rates up to $\frac{dz}{dt}$ = 5.0 cm s$^{-1}$), with an option to include a shape file describing where
erosion may occur e.g. to account for engineering structures such as check dams or sills, or natural
features such as bedrock, where significant erosion is not expected during one debris-flow event.
For comparison we also used a rate of $\frac{dz}{dt}$ = 1.25 cm s$^{-1}$ based on a lower rate from Berger et al.

289 (2011).

### 4. Erosion and entrainment: observations and modeling results

### 4.1. Erosion patterns and entrainment model calibration

The observed erosion patterns are the basis for calibrating the RAMMS model coefficents, in
particular the friction coefficients $\xi$ and $\mu$ are systematically adjusted in successive model runs,
until a satisfactory model result is achieved. The erosion pattern is derived by assessing the
difference between the digital elevation models. In both study areas, a measured erosion pattern
caused by one single debris flow event is not available. We therefore focus on the spatial




distribution of erosion and deposition, instead of attempting to exactly predict the spatial change
due to the debris flow process.
In the Meretschibach, the change in the DTM includes the erosion due to three debris flow events
which appear to have originated on an open-slope talus deposit (Figure 3A). The location of the
release area at the Meretschibach corresponds to the upper most visible erosion scar visible in the
DTM analysis and as described above includes the erosion due to three debris flow events between
July 17 to October 28, 2014 (Fig. 3A). Therefore, the release area was placed within the channel,
where up to 2.5 meters of erosion was observed (upper end of the blue polygon at about 1750 m
a.s.l. in Fig. 3A.). The location is just below a bedrock step intersecting the main channel at about
1800 m a.s.l. Further monitoring at the upper Bochtür subcatchment using interval cameras and
conducting field observations on the site itself confirmed that at least some of the debris flows most
likely initiated at this location.
We calibrated the RAMMS model using an initial block release volume of 10 $m^3$ which
corresponds to the channel depth of 1-2 m and a width of 2-4 m at this location. To keep the initial
volume within the channel and prevent unrealistic lateral outflow, no-flux boundaries were created
at the lateral sides of the initial landslide block. Within the middle and lower channel sections (Fig.
3A, blue polygon), the observed runout and relative erosion patterns can be best reproduced using
Voellmy friction parameters $\xi = 200$ $ms^{-2}$ and $\mu = 0.6$ (Fig. 3B2). The modeled velocities of
6-9 $ms^{-1}$ using $\xi = 200$ are plausible, although independent field data are not available for
comparison. The parameter combination $\xi = 200$ $ms^{-2}$ and $\mu = 0.7$ results in overbank flow along
both sides of the middle channel, which was not observed in the field (Fig. 3C2). There were
neither deposits outside of the channel nor were levees deposited along this entire channel reach
(Fig. 3A, blue polygon). In contrast, the erosion pattern using $\xi = 200$ $ms^{-2}$ and $\mu = 0.5$ resulted in
an even distribution of erosion along the entire channel length, which is inconsistent with the field
results which showed locations of deeper erosion depths (Fig. 3A). Within the normal range of the
$\xi$ parameter (Bartelt et al., 2013) the differences in flow and erosion patterns were small in
comparison to those resulting from variations in $\mu$, and are therefore not described herein. Hence,
the further model runs were conducted using the best-fit parameters $\xi = 200$ $ms^{-2}$ and $\mu = 0.6$ in the
sensitivity analyses described in subsequent sections.
In the Bondasca catchment, the differential elevation model includes both the rock avalanche
deposit (27 December 2011) and the erosion due to one debris-flow event (5 July 5, 2012) (Fig. 5).
The upper end of channel erosion is located just below a planar outcrop of bedrock (Fig. 4B)
corresponding to the likely location debris flow initiation zone (Fig. 5C). The surface runoff
channels along the west side of the wall and runoff across the wall surface (Fig. 4B) converge on
the sediments at the bottom of the rock wall (see pictures from 2014 in Fig. 5). This scenario
suggests a firehose-type debris-flow initiation (e.g. Godt and Coe, 2007). Hence, this location was
used for the runout modeling.




The observed erosion along the main debris flow channel (Fig. 5C) – resulting from the two debris
flow events in July 2012 – were used to calibrate the RAMMS model within the upper two thirds of
the study reach (Figure 4B, brown polygon) by varying the model parameters $\xi$ and $\mu$. The best fit
was found with the parameter combination $\xi = 400$ ms$^{-2}$ and $\mu = 0.3$. However, the observed
elevation change also includes secondary processes such as lateral bank collapse and the deposits
of debris-flow snouts and levees within the channel. Channel sections where the events eroded into
the deposits present can also be identified by the stratigraphy in the field.

## 4.2. Entrainment modeling and runout patterns

The runout of a (landslide-type) block release of 10 m$^3$, neglecting erosion (Fig. 6A) results in
maximum flow heights smaller than 0.5 m and the flow stops in the channel upstream of the
deposition zone. By contrast, including debris-flow erosion (Fig. 6B) leads to a more realistic flow
pattern consisting of flow within the channel reaching the deposition zone without any lateral
outflow. For comparison, if the total event volume ($\approx 1,555$ m$^3$) is released as a landslide and the
debris-flow is not allowed to erode the channel (Fig. 6C), the runout shows overbank flow along
the upper channel reaches below the initiation area. The last scenario is a typical example of how
debris-flow runout models are used when the total event volume is known. These results illustrate
the ability of the runout model to better predict the erosion pattern if the channel-bed erosion and
bulking process is included in the model.

## 4.3. Erosion model sensitivity testing

The results show that the total volume of eroded sediment, at both field sites, depends strongly on
the initial landslide volume. At both the Meretschibach and the Bondasca catchments, there is a
strong increase in the amount of sediment entrained and consequent increase in debris-flow volume
(Fig. 7) for relatively small increases of the initial landslide volume. At the Meretschibach
catchment, the erosion model – using the default maximum erosion rate $\frac{dz}{dt} = 2.5$ cm s$^{-1}$– shows the
highest sensitivity to the total erosion volume between 2 and 3 m$^3$ of initial block release (e.g.
initial landslide volume). Above 4-5 m$^3$ of initial block volume the increase of the total erosion
volume within the erosion domain remains approximately constant. The cause for the rapid
increase is related to the critical shear stress in the entrainment algorithm. Small initial landslides
do not generate enough shear stress to initiate erosion, whereas larger landslides can cause erosion
over the entire computational domain.
If we double the erosion rate to $\frac{dz}{dt} = 5.0$ cm s$^{-1}$ based on field estimates reported by Frank et al.,
2015 for the Spreitgraben catchment, a similar pattern is observed in the relationship between total
erosion volume as a function of initial release volume. However the erosion volumes are 3 to 5
times larger than the ones resulting from the default erosion rate at the same initial release volume.





In contrast, implementing only half the default maximum erosion rate ($\frac{dz}{dt}$ = 1.25 cm s$^{-1}$) for low
erosion scenarios decreases the sensitivity to initial volume in an analogous manner.
Similar trends in total erosion volume as a function of initial block release (landslide) volume are
observed at the Bondasca catchment. However, the model only starts to predict significant erosion
volumes for block releases exceeding 20 m$^3$, and the progressive increase in total erosion volume
as a function of initial block release volume is somewhat less steep. For the default erosion rate $\frac{dz}{dt}$
= 2.5 cm s$^{-1}$ (Frank et al., 2015), total erosion volumes increase most strongly between initial
volumes of 20 to 100 m$^3$. The topography at the Bondasca catchment is somewhat less steep and
more variable, which may help explain these differences. Doubling the default erosion rate at the
Bondasca catchment results in the onset of erosion for initial volumes between 20 and 30 m$^3$. When
reducing the default erosion rate to half of the default value, the erosion model depicts a somewhat
less sensitive reaction of the erosion model than using the default rate.
**5. Discussion**
The total erosion volumes observed in the sensitivity tests (Fig. 7) indicate a strong sensitivity to
block release volume (initial landslide volume) over a relatively narrow range of block release
volumes. This result is based on the assumption that the entire landslide fails instantaneously and
not progressively as a sequence of smaller landslides over a longer period of time. Information on
the style of initial landslide failure are not available for either field site, therefore we focus the
discussion on other factors related to the runout modeling. One striking difference between the two
field sites is that the size of the block release necessary to cause significant erosion is an order of
magnitude larger at the Bondasca site. The channel cross-sectional area where the flow travels and
therefore where the erosion model is active is different at the two field sites. The Meretschibach is
substantially steeper (50 to 65% vs. 15 to 35%). This results in larger shear stresses at the
Meretschibach for the same initial landslide thickness, because the shear-stress varies as the
product of initial release thickness, flow density, and channel slope. Other factors such as
differences in channel-bed roughness may also be important, however the Voellmy friction relation
within RAMMS does not explicitly consider channel-bed roughness.
In the RAMMS debris flow model, the development of the flow properties is controlled by the
Voellmy friction parameters ξ and μ (described in section 3.1) where ξ is the dominant control over
the flow velocities when the flow is moving rapidly and μ controls the runout distance. The ξ
parameter was found in this study to have a relatively small influence over the flow behavior in
comparison with the Coulomb friction term μ. The RAMMS manual (Bartelt et al., 2013) suggests
using the tangent of the fan slope as first estimate to determine μ. As described in the calibration
procedure (section 3.2), this corresponds to relative erosion patterns measured by differential DTM
analysis. Hence, we conclude that the tangent of the channel slope can be used as a first approach





to define parameter µ also for the erosion model, which was also found to be useful by Frank et al.
(2015) in the first application of the model.
Morphological effects influence the erosional behavior of the field data based erosion model. The
Bondasca channel is more variable in width and planform direction compared to the comparably
uniform and straight Meretschibach channel. This difference will cause larger spatial variability in
shear stress at Bondasca channel and therefore the channel will have a more variable onset of
debris flow erosion along the length of the channel. In the Bondasca catchment, the channel where
erosion takes place is significantly wider (4-10 m) than in the Meretschibach (1-3 m). On the one
hand, the flow can laterally spread more often in Bondasca than in the Meretschibach, thereby
locally reducing flow height, shear stresses and maximum potential erosion depth. On the other
hand, once the critical shear stress is exceeded, the potential erosion depth tends to increase more
rapidly in a narrow channel such as in the Meretschibach channel.
Another difference between the Meretschibach and the Bondasca channels is that the Bondasca
channel bed has a rougher surface with more scours holes, and larger blocks within the channel
which are similar in size to the nominal width of the channel. The model does not consider local
variations in erodibility due to the presence of large blocks, so local scour patterns in the field
around the large blocks are not present in the model results. Prancevic and Lamb, (2015a)
suggested that in rough mountain channels the large particles can be interlocked and hence more
stable. In contrast, local concentration of the flow between such large blocks may cause locally
very large shear stresses and corresponding large erosion rates. However, we do not have enough
information on the mobility of the large blocks, so this question cannot be addressed in more detail
herein.
The current version of the RAMMS model with erosion (version 1.6.45) does not adjust the
elevation of the bed when erosion occurs. The erosion can be subtracted from the initial DTM as a
post-processing step within the user interface, e.g. for modeling subsequent surges. This issue was
discussed at length by Frank et al (2015), and it can potentially complicate the interpretation of
erosion patterns resulting from multiple debris flows. Insufficient field data are available to help
constrain the events described herein.
Further assessment of the relation of the total erosion volumes depending on the initial volumes can
be made by calculating a bulking factor. The bulking factor BF is the ratio between the total
erosion volume $V_{ero}$ to the initial volume $V_{ini}$ :
$$BF = V_{ero}/V_{ini} \tag{8}$$
At the Meretschibach channel, the bulking factor is ≈ 200 when the erosion model using the default
erosion rate and an initial volume of 3 m$^3$ (Fig. 8). The BF reaches a peak $BF_p$ ≈ 300 at a release
volume of 4 m$^3$. It then drops to a BF ≈ 30 for an initial volume of 100 m$^3$. The model simulations
using the doubled default erosion rate show a bulking factor peak $BF_p$ ≈ 1,800 for an initial release





volume of 2 m$^3$; half the default erosion rate shifts this peak to 50 m$^3$ for the initial volume but the
corresponding peak bulking factor drops significantly down to ≈ 14.
The behavior of the bulking factor for the default erosion rate at the Bondasca catchment is
relatively smooth when compared to that at the Meretschibach. A peak bulking factor can be
identified somewhere between 200 and 500 m$^3$ but the value is lower in comparison (≈ 11) for the
default erosion rate. The doubled rate leads to a peak bulking factor BF$_p$ of ≈ 700 at a release
volume of 30 m$^3$. That is still large compared to examples in the literature (BF from 10 to 50
reported by Berti et al., 1999 and Vandine and Bovis, 2002). Nevertheless, a several hundred fold
increase of the debris flow volume due to bulking is plausible for extreme erosion cases. Larger
erosion rates might be expected for pycroclastic deposits (not present in the catchments described
herein) or due to the presence of very recent rock avalanche deposits which may contain firn-ice-
debris mixtures (e.g. Spreitgraben, Tobler et al., 2014; Frank et al., 2015). Large erodibilities may
be expected at the Bondasca catchment because the rock avalanche event occurred during winter
and may have contained significant amount of snow.
Due to the very long (≈ 4 km) and flat (≈ 15%) channel section in the middle segment of the
Bondasca catchment, the estimated deposition volumes (≈ 40,000 m$^3$) above the inlet of the
Bondasca river in the central valley are highly influenced by further erosional and depositional
processes along the channel.
**6. Conclusion**
Debris-flow runout predictions can be improved when considering the increase in flow volume
along the flow path. Using a recently-introduced empirical erosion algorithm within the RAMMS
2D runout model (Frank et al., 2015) we illustrate that runout patterns at the Meretschibach and
Bondasca catchments, in Switzerland, can be accurately modeled. When calibrated with field data,
the model produces more realistic runout patterns compared to simulations which do not consider
entrainment and bulking. In particular, we could show that even in very steep (≈ 60–65 %) and
narrow (4–6 m) torrent channels, lateral overflow – not observed in the field case – is prevented
when applying the entrainment model. However the model results can be quite sensitive to the
volume of the initial block release in the model which corresponds to the initial landslide volume.
The predicted erosion volumes are sensitive to the initial debris flow volume, with bulking factors
approaching 2000 predicted by the model, depending on the scenario considered. However, the
results are also sensitive to slope angle and channel morphology. The two field sites differ
substantially: the Meretschibach catchment is very steep with a straight and narrow channel,
whereas the Bondasca channel is less steep but morphologically more complex, yet the calibration
procedure is the same as for the standard RAMMS model which does not include the entrainment
process. The overall method presented herein is useful for case studies where sufficient data are





available to constrain the model results. However, more case studies have to be conducted to
develop a more comprehensive recommendation for modeling the runout of erosive debris flows in
natural terrain.
**Acknowledgements**
This project was partially supported by the CCES-TRAMM project. We are grateful to Christian
Huggel for helpful discussions and comments. We thank Martin Keiser of Amt für Wald und
Naturgefahren of Canton Graubünden for providing elevation data for the Bondasca catchment and
Ruedi Bösch, WSL,for the elevation data at the Meretschibach catchment.



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





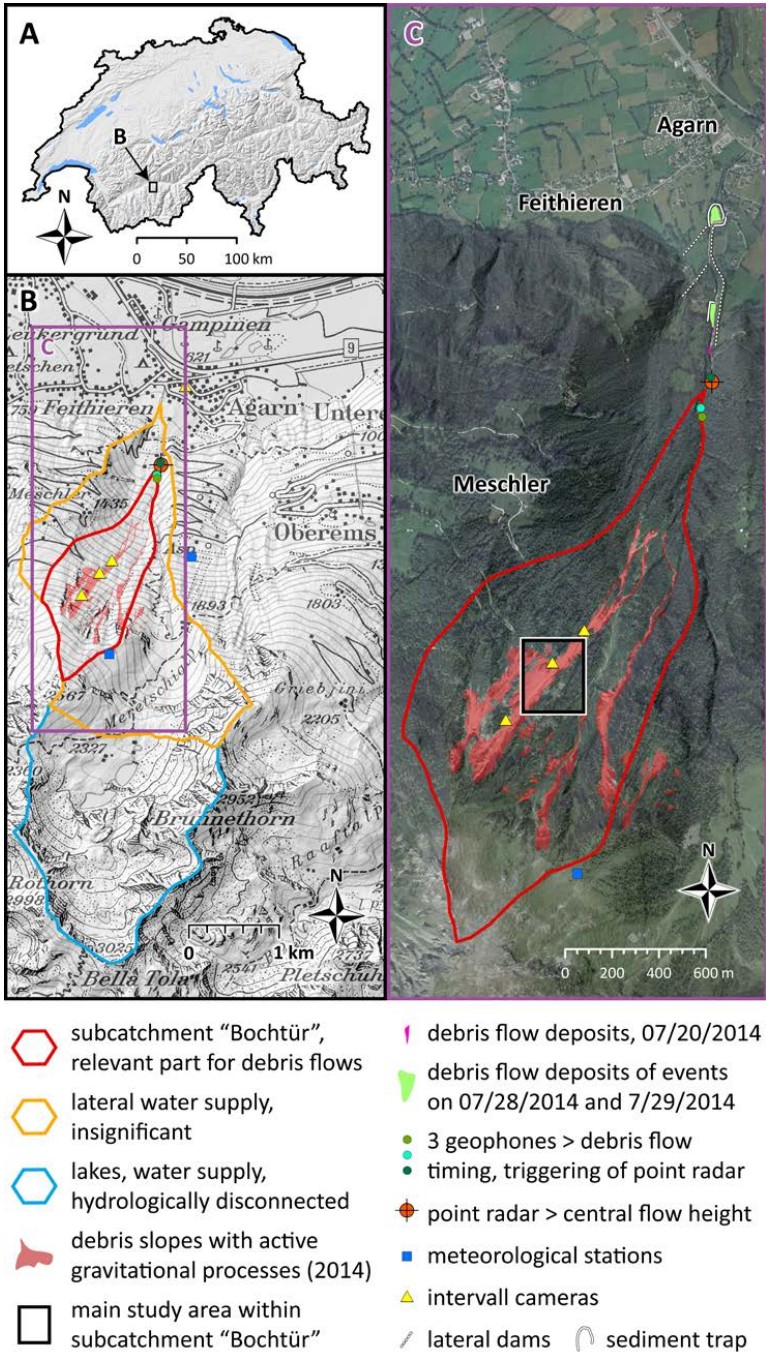

**Figure 1. A.** Location of the Meretschibach catchment in Southern Switzerland. **B.** Subcatchments
of the Meretschibach and locations of the instrumentation site and data available for the erosion
model analyses **C.** Initiation zone of the July 2014 events and camera positions. The main study




channel reach for the model testing is located in the middle part of "Bochtür" (black-white
retangle), swissimage©2014, swisstopo (5704 000 000) (2014).

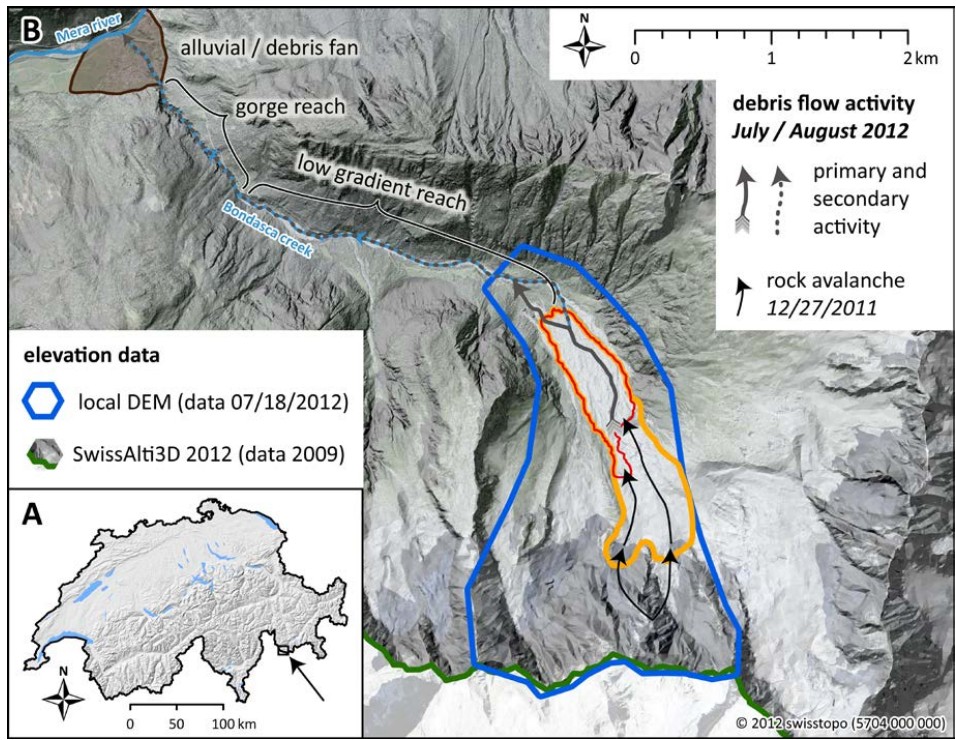


**Figure 2. A.** Location of the Bondasca catchment in south-eastern Switzerland close to the border
to Italy. **B.** Perimeter of the 27 December 2011 rock avalanche deposit, including the main
deposition area (yellow polygon) and the deposits lower-elevation deposits which have been
partially exposed to erosion by debris flows in 2012 (red polygon). The 2012 post-event digital
elevation model (lidar, blue polygon) is from 18 July 2012 (data courtesy of the Amt für Wald,
Canton Graubünden). Pre-event digital elevation model (lidar) for 2009 is from the SwissAlti3D
(version 2012) data set from swisstopo, ©2012, swisstopo (5704 000 000) . The grey solid arrow
indicates the main debris-flow channel formed in 2012.



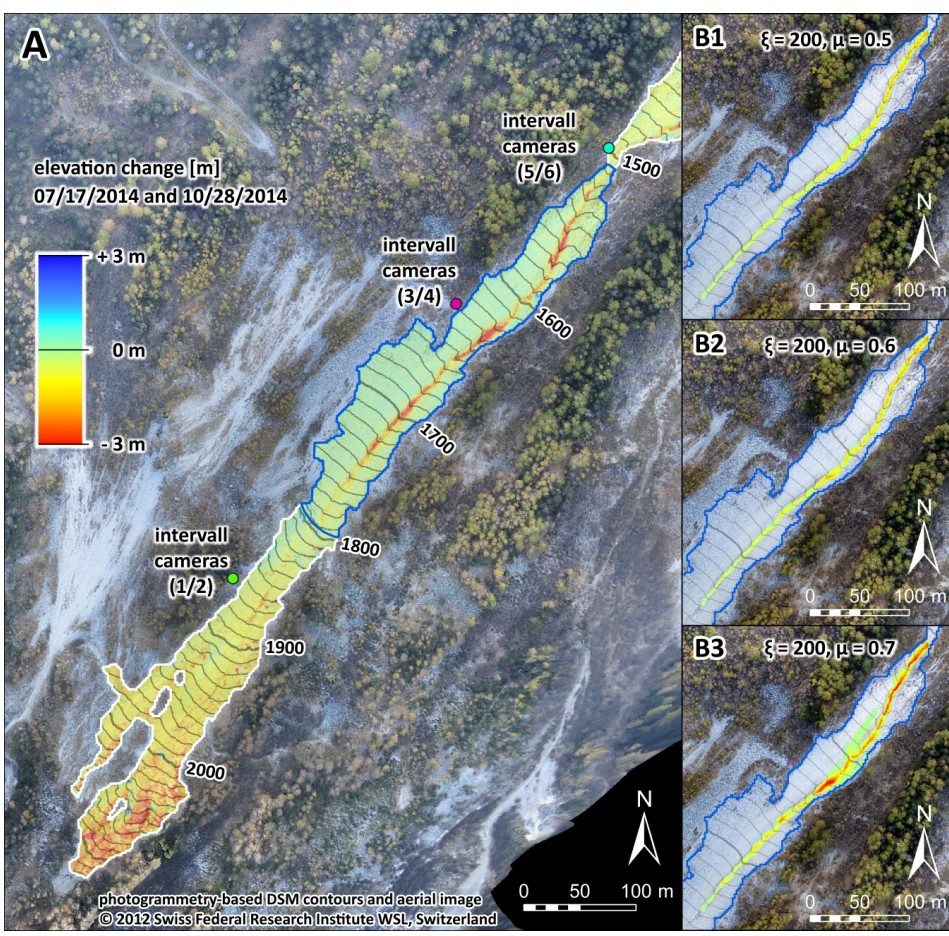

**Figure 3.** Calibration of modelled erosion patterns (**B1 to B3**) to the observed erosion depths (**A**) in the upper open debris slopes of the "Bochtür" catchment (Meretschibach) by varying values for the friction parameter μ. The blue polygon demarks the area where a differential DTM is available.





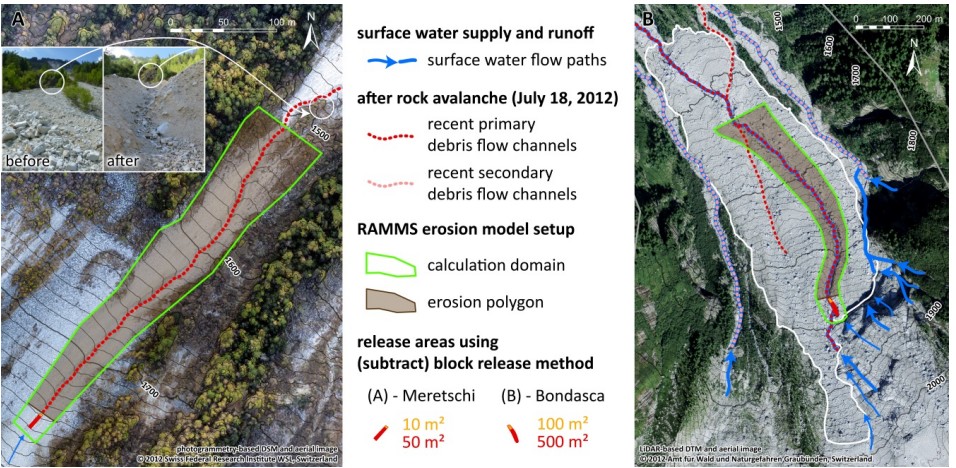

**Figure 4.** Erosion model configuration for the model simulations showing the initial block release
areas in the Meretschibach catchment (**A**) and the Bondasca catchment, Switzerland (**B**). The
hillslope is erodible within the brown shaded polygon.





**Figure 5.** Overview of rock avalanche deposits, subsequently formed debris flow channels, and the resulting overall elevation change in the Bondasca catchment (A, B). The elevation change map 2009 to 2012 (C) includes both the rock avalanche ( $\approx$ 1.5 Mio m$^3$ on 27 Dec. 2011) and the first two debris flow events (5 and 14 July 2012).






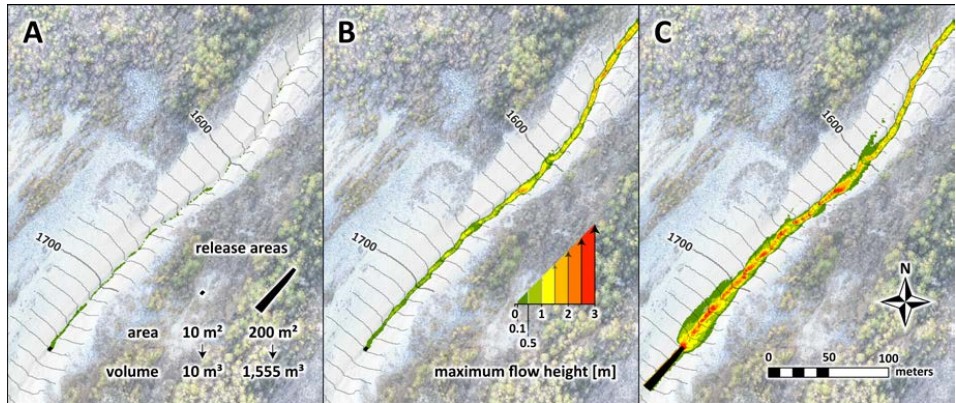

**Figure 6.** Comparison of runout patterns at "Bochtür" in the Meretschti catchment. The debris flow
modeling is conducted using a (subtract) block release volume of (**A**) 10 m³ and no-entrainment
modeling, of (**B**) 10 m³ and entrainment modeling as well as a total (subtract) block release volume
of (**C**) 1,555 m³ (sum of release and eroded volume from (**B**)) and no-entrainment modeling.





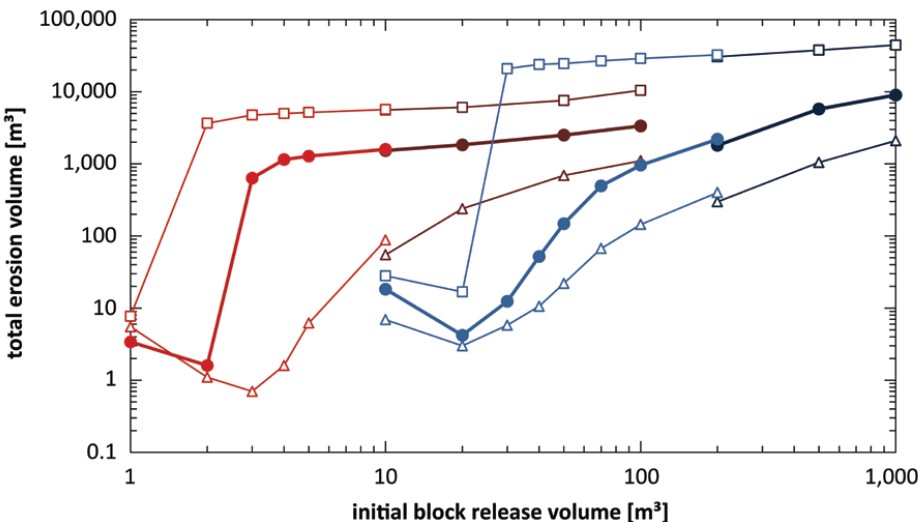

| | Meretschi | | Bondasca | |
|---|---|---|---|---|
| calibrated parameters | $\xi = 200\ m^2s^{-1},\ \mu = 0.60$ | | $\xi = 400\ m^2s^{-1},\ \mu = 0.30$ | |
| release areas | 10 m² | 50 m² | 100 m² | 500 m² |
| erosion rates (Frank et al., 2015) | 1.25 cm s⁻¹ | 2.5 cm s⁻¹ | | 5.0 cm s⁻¹ |

**Figure 7.** Sensitivity of modeled erosion volume to initial block release volume in the Meretschibach and in the Bondasca catchments.



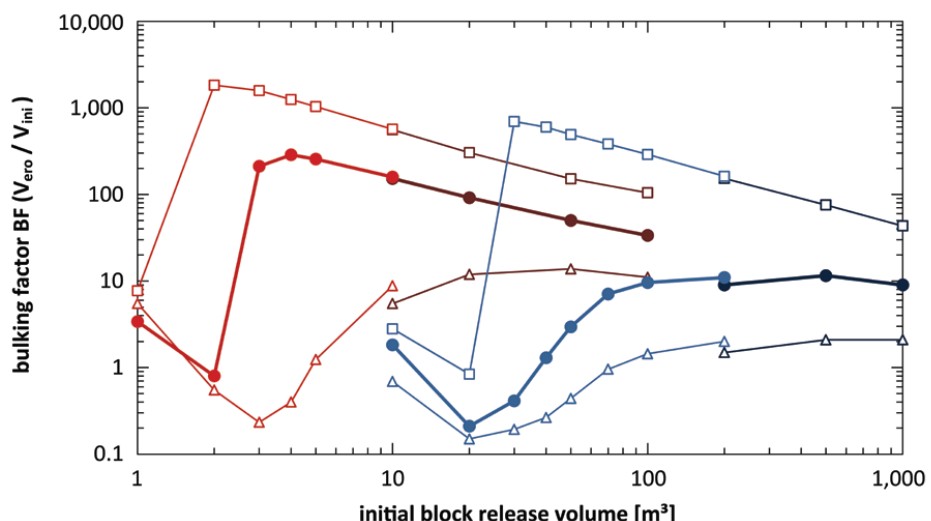

| | Meretschi | | Bondasca | |
|---|---|---|---|---|
| calibrated parameters | $\xi = 200$ m²s⁻¹, $\mu = 0.60$ | | $\xi = 400$ m²s⁻¹, $\mu = 0.30$ | |
| release areas | — 10 m² | — 50 m² | — 100 m² | — 500 m² |
| erosion rates (Frank et al., 2015) | ⃤ 1.25 cm s⁻¹ | ● 2.5 cm s⁻¹ | ⊡ 5.0 cm s⁻¹ | |


**Figure 8.** The bulking factor BF = $V_{ero}/V_{ini}$ of the modeled total erosion volume $V_{ero}$ [m³] to initial
block release volume $V_{ini}$ [m³] in the Meretschibach and Bondasca catchments.