# Peer review of "Debris flow modeling at Meretschibach and Bondasca catchment,"

_Natural Hazards and Earth System Sciences, 2016_

## Referee Comment (RC1) · Z. Han (Referee) · 21 Oct 2016

»General comments

This paper aims to simulate debris-flow process by considering bed erosion along the path. As erosion is a complex natural process and plays a very crucial role both in debris-flow dynamics, transportation, run out and deposition process, it is a very important research topic. To do so, this paper attempts to combine an empirical entrainment model which has been previously introduced by authors into the RAMMS model. The sensitivity of the developed model is tested by applying the model to two debris-flow events in Switzerland. The results show some interesting erosion and flow patterns.

Generally, this is a straight-forward development of the RAMMS model for the simulating bed erosion in debris flows. The paper is well illustrative and authoritative. The authors build upon their previous work and extend the RAMMS model to the erosion simulation. It is a major contribution and is sound. However, in my opinion, the major limitation of this paper is that the described debris-flow entrainment model is rather sensitive to the empirical coefficients, and these coefficients are not well illustrated in the paper. Indeed, the authors show us a sensitivity analysis of erosion volume to the initial volume, erosion rate and calibrated parameters $\xi$ and u in Fig.7. As they mention, the value of these parameters are suggested by the previous study in the same region, e.g., the erosion rate dz/dt = 2.5cm/s. However, the rational range of these parameters may be different in other regions, there is a need to explain how to determine these parameters. The paper could be improved and made more accessible by further exploring these empirical parameters. I recommend this paper for publication after major revisions.

»Specific comments

Page 6, 195-196. The authors mention that the slope angle $\Phi$ in the deceleration term $S_f$ is similar to the internal friction angle of the material. Does it mean that $S_f$ will be the same when at a steep slope and a gentle slope? Please check it.

Page 7, 221-222. The critical shear stress $\tau_c$ determines the maximum potential erosion depth $e_m$, the erosion will not be existed at the area where $\tau < \tau_c$. For this reason, the critical shear stress is a key parameter for controlling the shape of erosion area and erosion depth. But the authors superficially use an empirical value 1kPa in the paper, and no sensitivity analysis is made. It seems that they could simply test and provide results on how sensitive the simulation is to the choice of the critical shear stress $\tau_c$.

Page 11, 360. The total erosion volume remains approximately constant when the initial volume exceeds a certain value. How to explain this phenomenon? Is this because the maximum erosion depth $e_m$ is reached as controlled by the critical shear stress

$\tau$c=1kPa? As such, it seems that the choice of $\tau$c as a model parameter should be discussed to a greater degree, especially if you want your method to be used more widely on debris flows of varying properties.

Page 12, 396-397. As I see in Fig.3, there is no significant difference of runout distance in B2 (u=0.6) and B3 (u=0.7). Please check the sentence "u controls the runout distance".

Page 27, 675. The total erosion volume in both cases show an abrupt decrease, and then a significant increase with the initial release volume, i.e., 1-2m3 in Meretschi and 10-20m3 in Bondasca. Is there any rational explanation on it?

---

## Referee Comment (RC2) · Anonymous Referee #2 · 2 Jan 2017

[revised manuscript text omitted]

channel reach for the model testing is located in the middle part of "Bochtür" (black-white retangle), swissimage©2014, swisstopo (5704 000 000) (2014).

[Figure]

**Figure 2. A.** Location of the Bondasca catchment in south-eastern Switzerland close to the border to Italy. **B.** Perimeter of the 27 December 2011 rock avalanche deposit, including the main deposition area (yellow polygon) and the deposits lower-elevation deposits which have been partially exposed to erosion by debris flows in 2012 (red polygon). The 2012 post-event digital elevation model (lidar, blue polygon) is from 18 July 2012 (data courtesy of the Amt für Wald,

Canton Graubünden). Pre-event digital elevation model (lidar) for 2009 is from the SwissAlti3D

(version 2012) data set from swisstopo, ©2012, swisstopo (5704 000 000) . The grey solid arrow indicates the main debris-flow channel formed in 2012.

[Figure]

[Figure]

[Figure]

**Figure 3.** Calibration of modelled erosion patterns (**B1 to B3**) to the observed erosion depths (**A**) in
the upper open debris slopes of the "Bochtür" catchment (Meretschibach) by varying values for the
friction parameter μ. The blue polygon demarks the area where a differential DTM is available.

[Figure]

[Figure]

[Figure]

**Figure 4.** Erosion model configuration for the model simulations showing the initial block release areas in the Meretschibach catchment (**A**) and the Bondasca catchment, Switzerland (**B**). The hillslope is erodible within the brown shaded polygon.

[Figure]

[Figure]

**Figure 5.** Overview of rock avalanche deposits, subsequently formed debris flow channels, and the resulting overall elevation change in the Bondasca catchment (A, B). The elevation change map 2009 to 2012 (C) includes both the rock avalanche ($\approx$ 1.5 Mio m$^3$ on 27 Dec. 2011) and the first two debris flow events (5 and 14 July 2012).

[Figure]

[Figure]

[Figure]

**Figure 6.** Comparison of runout patterns at "Bochtür" in the Meretschti catchment. The debris flow modeling is conducted using a (subtract) block release volume of (**A**) 10 m³ and no-entrainment modeling, of (**B**) 10 m³ and entrainment modeling as well as a total (subtract) block release volume of (**C**) 1,555 m³ (sum of release and eroded volume from (**B**)) and no-entrainment modeling.

[Figure]

[Figure]

|  | **Meretschi** | | **Bondasca** | |
|---|---|---|---|---|
| **calibrated parameters** | $\xi = 200\ m^2 s^{-1}$, $\mu = 0.60$ | | $\xi = 400\ m^2 s^{-1}$, $\mu = 0.30$ | |
| **release areas** | — $10\ m^2$ | — $50\ m^2$ | — $100\ m^2$ | — $500\ m^2$ |
| **erosion rates** **(Frank et al., 2015)** | —△— $1.25\ cm\ s^{-1}$ | —●— $2.5\ cm\ s^{-1}$ | —☐— $5.0\ cm\ s^{-1}$ | |

**Figure 7.** Sensitivity of modeled erosion volume to initial block release volume in the
Meretschibach and in the Bondasca catchments.

[Figure]

[Figure]

[Figure]

**Figure 8.** The bulking factor BF = $V_{ero}/V_{ini}$ of the modeled total erosion volume $V_{ero}$ [m$^3$] to initial block release volume $V_{ini}$ [m$^3$] in the Meretschibach and Bondasca catchments.

---

## Author Comment (AC2) · 8 Feb 2017

**Reviewer 2: Anonymous**

**General comments**

The paper deals with bed entrainment for debris flows in Switzerland using numerical modelling. The topic is of interest for the Journal and the specific issues of this paper are relevant to scientists and practitioners. Some (mandatory) changes are required to improve the paper before acceptance. The list of specific comments and suggestions is given in the attached file.

**Authors response:** We are grateful for the helpful specific comments, especially literature citations which were not cited in the last version of the manuscript. These comments should substantially improve the manuscript. Please see our responses to the specific suggestions below.

**Specific comments**

**Reviewer 2: Page 1, lines 23-24.**

why this choice. Basal friction and bed entrainment are interplaying in natural processes. Why separate calibration?

**Authors:** We decided to first calibrate the runout of the model based on the total volume of the event and the runout distance, and then work with smaller initial volumes, then including the erosion algorithm, to refine the results. Our goal was to avoid a time-intensive iterative procedure, especially for the benefit of practitioners who generally do not have time to go through a long calibration process. However the model could also be calibrated by starting with small landslide volumes, so this is just a statement of how we performed the calibration.

**Reviewer 2: Page 2, line 34.**

you mean rheology?

**Authors:** This sentence would be better stated as follows: "Sediment erosion caused by debris flows causes flow bulking (in our case an increase in flow mass; Iverson 1997) which strongly influences the runout behavior of debris flows." We suggest to change it to clarify this.

**Reviewer 2: Page 2, line 38.**

quote also works of:

- Cascini et al. (2106) Eng Geol

- Cuomo et al. (2016) Eng Geol

- Cuomo et al., (2014) Canadian Geotechnical Journal where bed entrainment is discussed as far as its spatial-temporal variation, and its interplay with rheology

**Authors:** Thank you for pointing out this additional literature, which we did not initially consider for this manuscript. However our focus in not on the rheology of the flow or changes in the rheology as a consequence of entrainment. As stated in the manuscript, we use the Voellmy friction relation and we do not adjust the Voellmy friction coefficients as a function of flow properties.

However we propose including this as a discussion point, where we will be able to cite some of these publications.

**Reviewer 2: Page 2, line 45.** what is this? bed entrainment? you may also call erosion. But bulking process is hard to understand and not common in international literature.

**Authors:** The term "bulking" is commonly used in the literature to describe the increase in mass of a debris flow along the flow path ,e.g. see Iverson, R. M.: The Physics of Debris Flows, Reviews of

Geophysics, 35, 245-296, 1997. doi: 10.1029/97RG00426, 1997, for a clear explanation in a paper which is very widely cited by debris-flow and landslide researchers throughout the world. A quick search on an academic search engine also indicates that "bulking" is commonly used in the debris- flow literature by authors from many countries outside of Switzerland, so we respectfully disagree with Reviewer 2 on this point. We realize that it may have other meanings in other academic disciplines, so we propose that we clarify the terms like this in the next version of the manuscript.

**Reviewer 2: Page 2, line 55.**

is there any difference?

**Authors:** Erosion removes sediment from the channel bed, bulking describes the increase in size (mass) of the flow, so the two terms are closely related but not interchangeable. As stated above, we will, in the next version, provide definitions of the terms.

**Reviewer 2: Page 2, lines 57-61.**

there are cases where neglecting erosion one may obtain unsafe future scenarios, as bed entrainment change the propagation pattern, and thus influence the global behaviour of the landslide. This is especially true for debris avalanches (not channelised). However, also for debris flows, including the entrainment helps obtaining better model estimates. See, for instance Cascini et al. 2014 Geomorphology

**Authors:** Thank you for pointing out this paper, which we will consider citing for the next version of the paper. We agree that including entrainment may help users to obtain more accurate predictions.

**Reviewer 2: Page 2, lines 73-74.**

add models by Pastor et al.. You may find applications in previous works of Cuomo et al.

**Authors:** Thank you for pointing out these additional papers, which we will cite, if appropriate, in for the next version of the manuscript.

**Reviewer 2: Page 2, lines 76-79. Also line 165 (which does not have a comment, just a**

**highlight).**

are you using erosion, entrainment and bulking with the same content?

**Authors:** It is not clear to us if this comment is about the terminology or the differences in the bulk properties of the flow vs. the channel bed, so we will address both comments:

A. In our case, the bulking (increase in mass of the flow) produced by entrainment (the process described in the model which specifies how fast and where the additional sediment enters the debris flow) should be clear (also see our comments above regarding terminology). Net entrainment of sediment (erosion – deposition) results in net erosion of the channel bed (a decrease in the elevation of the channel bed), which can then be characterized in a spatial sense with a description of a pattern.

B. Although it is possible to specify a different mass density for the sediment that is entrained from the channel bed, to a first approximation the mass densities of the two are similar, at least in torrents which experience frequent debris flows. In more detail, the degrees of sorting and ranges of grain sizes in both the flow deposits and the channel bed are fairly similar. However the model accounts for differing densities, if such values are available.

**Reviewer 2: Page 6, line 192.**

turbolent factor. And, it is does not depend on v^2. rephrase the whole sentence.

**Authors:** Thank you for pointing out that this is not clear to you, we propose that we re-write the sentence in question.

**Reviewer 2: Page 7, line 220.**

??

**Authors:** Thank you for pointing out the error in the reference number of the equation, we will fix that in the next version of the manuscript (it should be Eq. 6).

**Reviewer 2: Page 7, line 221.**

from where this value? / from where?

**Authors:** These values were described by Frank et al. (2015), however upon re-reading the paragraph above Equation 6, we realize that we should add more details in the next version of the manuscript. Additionally, we propose adding "Frank et al. (2015)" at the end of the sentence to make the origin more clear to the reader.

**Reviewer 2: Page 7, lines 229-230.**

check numbering of eqs

**Authors:** Thank you for pointing out the error in the reference to the equation, we will correct and verify all equation numbers when preparing the next version of the manuscript.

**Reviewer 2: Page 9, line 281.**

-2.5 ?

**Authors:** We agree with your suggestion we will also change the value to SI units, so -0.025 m/s, also for other occurrences of $\frac{dz}{dt}$ values in the manuscript.

**Reviewer 2: Page 10, line 314.**

how this was fixed?

**Authors:** The parameter $\xi$ was determined by varying it within the range proposed by the developers of the RAMMS model ($\xi$ = 100, 200, 400) and inspecting the results. The only realistic velocities (in the steep ($\approx$60%) study reach of the Meretschibach channel) are obtained using $\xi$ = 200 when combined with the variation of parameter $\mu$ (= 0.5, 0.6, 0.7). This is explained in the manuscript on page 10, lines 312-316. However to ensure that this is clear, we propose adding a sentence to clarify this procedure.

**Reviewer 2: Page 13, line 432.**

Alternative, but related definition is that of Hungr, i.e. landslide growth rate = Vfinal / Vinitial

**Authors:** Thank you for pointing out Hungr's definition. We will verify which metric are used in the other papers which we reference, and for the next version of the manuscript we will choose the most suitable metric (as well as cite Hungr's landslide growth factor).

---

## Author Response (AR1)

*Response file on* **"Debris flow modeling at Meretschibach and Bondasca catchments,**

**Switzerland: sensitivity testing of field data-based erosion model"** by F. Frank, B.W. McArdell,

N. Oggier, P. Baer, M. Christen and A. Vieli florian.frank@wsl.ch

**Reviewer 1:** Z. Han

**Reviewer 2:** Anonymous

**Editor:** M. Keiler

**(A) Comments from referees/public**

**(B) Authors response**

**(C) Authors changes**

**General comment by the authors**

In addition to the comments from the Editor and Reviewers, a few additional minor changes to the text have also been made for clarity. These changes are also visible in the document highlighting the changes made to the manuscript. A few formal errors (e.g. typos, etc.) were also fixed.

**PLEASE NOTE:** The page/line(s) references provided by **(A) reviewers and editor** and **(B)**

**Response** still refer to first submission manuscript.

But all page/line(s) references given by the authors (**"(C) Changes"**) in this document refer to the new resubmission manuscript (which highlights track changes in RED and text moved in GREEN).

**General comments by Reviewer 1**

**(A) Reviewer 1:** This paper aims to simulate debris-flow process by considering bed erosion along the path. As erosion is a complex natural process and plays a very crucial role both in debris-flow dynamics, transportation, run out and deposition process, it is a very important research topic. To do so, this paper attempts to combine an empirical entrainment model which has been previously introduced by authors into the RAMMS model. The sensitivity of the developed model is tested by applying the model to two debris-flow events in Switzerland. The results show some interesting erosion and flow patterns.

**(B) Response:** We are grateful for the thorough reading of the manuscript and the helpful review, which we think will substantially improve the manuscript. We address both the general and specific comments below.

**(C) Changes:** Please see further general comments of Reviewer 1 below and "Specific comments".

**(A) Reviewer 1:** Generally, this is a straight-forward development of the RAMMS model for the simulating bed erosion in debris flows. The paper is well illustrative and authoritative. The authors build upon their previous work and extend the RAMMS model to the erosion simulation. It is a major contribution and is sound. However, in my opinion, the major limitation of this paper is that the described debris-flow entrainment model is rather sensitive to the empirical coefficients, and these coefficients are not well illustrated in the paper. Indeed, the authors show us a sensitivity analysis of erosion volume to the initial volume, erosion rate and calibrated parameters and in Fig.7. As they mention, the value of these parameters are suggested by the previous study in the same region, e.g., the erosion rate $dz/dt = 2.5$ cm/s. However, the rational range of these parameters may be different in other regions, there is a need to explain how to determine these parameters. The paper could be improved and made more accessible by further exploring these empirical parameters. I recommend this paper for publication after major revisions.

**(B) Response:** Yes, we agree that the results of the model are quite sensitive to the empirical coefficients and the initial conditions, which is the main focus of the manuscript. The uncertainties in the field data (e.g. initial flow volume, volume of eroded sediment, magnitude of erosion) are generally fairly large. It may be by chance that the same coefficients deliver plausible results at three different debris-flow sites in the Swiss Alps when in fact it may be possible to refine the coefficients in cases where more precise field data are available. For this reason, all of the coefficients can be adjusted when more or better results become available. We intend to edit the manuscript to make this point more clear. To further explore the influence of the parameter combinations, we also intend to highlight the inherent feedback in the model, whereby a rapid erosion rate results in an increase in flow depth leading to larger shear stresses and then to even larger potential erosion depths. This potentially explains the very rapid growth of debris flows, which is has been observed in some natural field cases and also in laboratory experiments involving realistic debris-flow sediments (e.g. Logan & Iverson, 2007, Video documentation of experiments at the USGS Debris-flow flume 1992-2006, U.S. Geological Survey Open-File Report 2007-1315).

**(C) Changes:** Editing of the manuscript was processed in sections 4.2. (page 12) and 5. (discussion, starting at page 14) to make the points mentioned above more clear. The reference list has been updated accordingly.

**General comments by Reviewer 2**

**(A) Reviewer 2:** The paper deals with bed entrainment for debris flows in Switzerland using numerical modelling. The topic is of interest for the Journal and the specific issues of this paper are relevant to scientists and practitioners. Some (mandatory) changes are required to improve the paper before acceptance. The list of specific comments and suggestions is given in the attached file.

**(B) Response:** We are grateful for the helpful specific comments, especially literature citations
which were not cited in the last version of the manuscript. These comments should substantially
improve the manuscript. Please see our responses to the specific suggestions below.
**(C) Changes:** Please see "Specific comments".

**Specific comments by Reviewer 1**

**(A) Reviewer 1: Page 6, 195-196.** The authors mention that the slope angle φ in the deceleration term $S_f$ is similar to the internal friction angle of the material. Does it mean that $S_f$ will be the same when at a steep slope and a gentle slope? Please check it.

**(B) Response:** Yes, it is true that the slope angle is similar to the angle of internal friction at this slope. However we do not imply that the value of $S_f$ in the Voellmy friction relation is the same as the value of internal friction in general. We intend to remove this comment to avoid confusion for readers, because the Voellmy friction angle is typically selected based on other criteria.

**(C) Changes:** The comment in parenthesis "(approximately similar to the internal friction angle of the material)" was removed (page 8, lines 208-209).

**(A) Reviewer 1: Page 7, 221-222.** The critical shear stress $\tau_c$ determines the maximum potential erosion depth $e_m$, the erosion will not be existed at the area where $\tau < \tau_c$. For this reason, the critical shear stress is a key parameter for controlling the shape of erosion area and erosion depth. But the authors superficially use an empirical value 1 kPa in the paper, and no sensitivity analysis is made. It seems that they could simply test and provide results on how sensitive the simulation is to the choice of the critical shear stress $\tau_c$.

**(B) Response:** Yes, we did not include results for the sensitivity analysis regarding the value of the critical shear stress, because the influence is generally much smaller and the range of critical shear stress values is small. We disagree that our choice and use of 1 kPa is superficial, because this value based on indirect observations by Schürch et al. (2011, cited in the original manuscript) and it serves a general purpose of describing that torrent channel beds are typically not eroded by small debris flows. We do not know the precise value in the field at any field site however a value near 1 gives a good fit to the data set we used for that analysis. Using a Shields's criteria for critical shear stress from river engineering, we find a critical shear stress which is smaller than 1, depending on the grain size on the channel bed. We discussed the issue of different critical shear stresses in our first erosion model application at the Spreitgraben catchment (Frank et al., 2015). Therein, we described smaller debris flood events which produced about 4-5 kPa of shear stress but did not show any significant erosion in the channel bed, i.e. suggesting that the critical shear stress $\tau_c$ may be somewhat larger in the Spreitgraben than in the Illgraben channel.

We propose inserting a paragraph discussing this issue in more detail, especially noting that the value is close to zero, or could be set to zero if field evidence indicates that erosion is always expected. However we will gladly include such a plot as an additional figure if this is desired by the editorial staff.

**(C) Changes:** We performed additional model runs to assess the sensitivity of the model results to the critical shear stress value. We also changed the comparison from the *bulking factor* to *volume growth* (Hungr et al., 2005). For this assessment, we selected the Meretschibach catchment because it has a simple single-channel morphology channel geometry and therefore serves as a clear case for illustration. The results are presented in the new Figure 8. The former Figure 8 is now Figure 9

- which we also updated to show results as volume growth instead of using the bulking factor BF.

The values for BF have all been recalculated as VG values and the manuscript has been updated accordingly. We added a short description of the results in section 4.3 (pages 13/14, lines 402-

411)., introduced the VG equation (Eq. 8, moved to section 4.3, page 13, line 407), and now we discuss the results as presented in Fig. 8 in a new paragraph in the middle of the discussion section (page 16).

**(A) Reviewer 1: Page 11, 360.** The total erosion volume remains approximately constant when the initial volume exceeds a certain value. How to explain this phenomenon? Is this because the maximum erosion depth $e_m$ is reached as controlled by the critical shear stress $\tau_c = 1$ kPa? As such, it seems that the choice of $\tau_c$ as a model parameter should be discussed to a greater degree, especially if you want your method to be used more widely on debris flows of varying properties.

**(B) Response:** In fact, the volume continues to increase (the y-axis is a logarithmic scale). The maximum erosion depth $e_m$ may be limiting – however $e_m$ also increases due to increasing maximum flow heights (see also Fig. 3 in Frank et al., 2015) when systematically enlarging the initial release volumes in the sensitivity analysis. Again, the model is insensitive to the value of critical shear stress once that value is exceeded (please refer to our comment above).

**(C) Changes:** See changes as described in the comment above. We added the new Figure 8

(sensitivity analysis for $\tau_c$ ). Changes were made in section 4.3 to describe the data presented in the new Figure 8 which we discuss in a new paragraph in the middle of our discussion (page 16).

**(A) Reviewer 1: Page 12, 396-397.** As I see in Fig.3, there is no significant difference of runout distance in B2 (µ=0.6) and B3 (µ=0.7). Please check the sentence "µ controls the runout distance".

**(B) Response:** The sentence "µ controls the runout distance" is consistent with the Voellmy friction relation as used in runout models such as RAMMS. It is not a result from this present study. We will provide a suitable literature citation for that statement and adjust the wording to ensure that readers do not see this as a result of this project. Perhaps the problem lies in the illustration of the modeling results for three µ values. The calculation domain, where the software was used, was limited in spatial extent to the area where we have differential DTM data for comparison (e.g. the blue polygon in Fig. 3A), so we actually do not show the final runout distance.

In the process of answering this comment, we noticed that the blue polygon is drawn inconsistently in figure 3B1-B3, but it is drawn correctly in Fig. 3A. We will also correct this error in the final manuscript.

**(C) Changes:** We now provide a literature citation "(Bartelt et al., 2013)" for the statement "µ

controls the runout distance (page 14, line 429). We've also adjusted the wording in this paragraph of the discussion chapter to ensure that readers do not see this as a result of this project (page 14, lines 431-435). We also adapted the inconsistently drawn blue polygons in figure 3B1-B3 to be consistent with the correctly drawn blue polygon in Fig. 3A.

**(A) Reviewer 1: Page 27, 675**. The total erosion volume in both cases show an abrupt decrease, and then a significant increase with the initial release volume, i.e., 1-2 $m^3$ in Meretschi and 10-20

$m^3$ in Bondasca. Is there any rational explanation on it?

**(B) Response:** When comparing erosion depths as modeled using 10 vs. 20 $m^3$ as the initial volume in the Bondasca case e.g., we observed that the model run using 20 $m^3$ is large enough that part of the flow enters a secondary channel. The volume of the flow, then divided among two channels, causes a reduction in flow depth and a consequent decrease in shear stress, resulting in smaller erosion depths and therefore smaller erosion volumes. When using an initial volume of 10

$m^3$ then the flow fully stays in the main channel. We propose adding a small discussion paragraph explaining this issue.

**(C) Changes:** We added a brief discussion paragraph explaining this issue (page 17, lines 517-

525).

**Specific comments by Reviewer 2**

**(A) Reviewer 2: Page 1, lines 23-24.**

why this choice. Basal friction and bed entrainment are interplaying in natural processes. Why separate calibration?

**(B) Response:** We decided to first calibrate the runout of the model based on the total volume of the event and the runout distance, and then work with smaller initial volumes, then including the erosion algorithm, to refine the results. Our goal was to avoid a time-intensive iterative procedure, especially for the benefit of practitioners who generally do not have time to go through a long calibration process. However the model could also be calibrated by starting with small landslide volumes, so this is just a statement of how we performed the calibration.

**(C) Changes:** We added a few words at this location in the abstract section to further clarify this (page 1, lines 24-27).

**(A) Reviewer 2: Page 2, line 34.**

you mean rheology?

**(B) Response:** This sentence would be better stated as follows: "Sediment erosion caused by debris flows causes flow bulking (in our case an increase in flow mass; Iverson 1997) which strongly influences the runout behavior of debris flows." We suggest to change it to clarify this.

**(C) Changes:** We changed the first sentence of the "Introduction" chapter as suggested (page 3, lines 39-41).

**(A) Reviewer 2: Page 2, line 38.**

quote also works of:

- Cascini et al. (2106) Eng Geol

- Cuomo et al. (2016) Eng Geol

- Cuomo et al., (2014) Canadian Geotechnical Journal where bed entrainment is discussed as far as its spatial-temporal variation, and its interplay with rheology

**(B) Response:** Thank you for pointing out this additional literature, which we did not initially consider for this manuscript. However our focus in not on the rheology of the flow or changes in the rheology as a consequence of entrainment. As stated in the manuscript, we use the Voellmy friction relation and we do not adjust the Voellmy friction coefficients as a function of flow properties. However we propose including this as a discussion point, where we will be able to cite some of these publications.

**(C) Changes:** We added this as a short discussion point and cite some of the publications in the paragraph which already described how morphological effects influence the erosional behavior of the model (page 15, lines 462-466). The reference list has been updated accordingly.

**(A) Reviewer 2: Page 2, line 45.** what is this? bed entrainment? you may also call erosion. But
bulking process is hard to understand and not common in international literature.
**(B) Response:** The term "bulking" is commonly used in the literature to describe the increase in
mass of a debris flow along the flow path ,e.g. see Iverson, R. M.: The Physics of Debris Flows,
Reviews of Geophysics, 35, 245-296, 1997. doi: 10.1029/97RG00426, 1997, for a clear
explanation in a paper which is very widely cited by debris-flow and landslide researchers
throughout the world. A quick search on an academic search engine also indicates that "bulking" is
commonly used in the debris-flow literature by authors from many countries outside of
Switzerland, so we respectfully disagree with Reviewer 2 on this point. We realize that it may have
other meanings in other academic disciplines, so we propose that we clarify the terms like this in
the next version of the manuscript.
**(C) Changes:** Further clarification of the term *bulking* is made by "(in our case an increase in flow
mass, e.g. Iverson, 1997)" in the first sentence of the Introduction section (page 3, lines 39-41).
Additional definitions of the terms *erosion*, *entrainment* and *bulking* are given in the following
sentences (page 4, lines 41-45). Throughout the rest of the manuscript, we made sure that we use
those three defined terms consistently.

**(A) Reviewer 2: Page 2, line 55.**
is there any difference?
**(B) Response:** Erosion removes sediment from the channel bed, bulking describes the increase in
size (mass) of the flow, so the two terms are closely related but not interchangeable. As stated
above, we will, in the next version, provide definitions of the terms.
**(C) Changes:** These changes have been addressed in our response on line number 202 of this
document.

**(A) Reviewer 2: Page 2, lines 57-61.**
there are cases where neglecting erosion one may obtain unsafe future scenarios, as bed
entrainment change the propagation pattern, and thus influence the global behaviour of the
landslide. This is especially true for debris avalanches (not channelised). However, also for debris
flows, including the entrainment helps obtaining better model estimates. See, for instance Cascini
et al. 2014 Geomorphology
**(B) Response:** Thank you for pointing out this paper, which we will consider citing for the next
version of the paper. We agree that including entrainment may help users to obtain more accurate
predictions.
**(C) Changes:** We cited Cascini et al. (2014) in the introduction section (page 4, line 77) and the
reference list has been updated accordingly.

**(A) Reviewer 2: Page 2, lines 73-74.**

add models by Pastor et al.. You may find applications in previous works of Cuomo et al.

**(B) Response:** Thank you for pointing out these additional papers, which we will cite, if appropriate, in for the next version of the manuscript.

**(C) Changes:** We cited Pastor et al. (2009) there (page 4, line 75) and the reference list has been updated accordingly.

**(A) Reviewer 2: Page 2, lines 76-79. Also line 165 (which does not have a comment, just a**

**highlight).**

are you using erosion, entrainment and bulking with the same content?

**(B) Response:** It is not clear to us if this comment is about the terminology or the differences in the bulk properties of the flow vs. the channel bed, so we will address both comments:

(1). In our case, the bulking (increase in mass of the flow) produced by entrainment (the process described in the model which specifies how fast and where the additional sediment enters the debris flow) should be clear (also see our comments above regarding terminology). Net entrainment of sediment (erosion – deposition) results in net erosion of the channel bed (a decrease in the elevation of the channel bed), which can then be characterized in a spatial sense with a description of a pattern.

(2). Although it is possible to specify a different mass density for the sediment that is entrained from the channel bed, to a first approximation the mass densities of the two are similar, at least in torrents which experience frequent debris flows. In more detail, the degrees of sorting and ranges of grain sizes in both the flow deposits and the channel bed are fairly similar. However the model accounts for differing densities, if such values are available.

**(C) Changes:** These changes have been addressed in our response on line number 202 of this document.

**(A) Reviewer 2: Page 6, line 192.**

turbolent factor. And, it is does not depend on v^2. rephrase the whole sentence.

**(B) Response:** Thank you for pointing out that this is not clear to you, we propose that we re-write the sentence in question.

**(C) Changes:** By revisiting the referenced RAMMS papers (Christen et al., 2010; Christen et al.,

2012; Bartelt et al., 2013), we re-wrote the entire sentence (page 7, line 202-206).

**(A) Reviewer 2: Page 7, line 220.** ??

**(B) Response:** Thank you for pointing out the error in the reference number of the equation, we will fix that in the next version of the manuscript (it should be Eq. 6).

**(C) Changes:** Error fixed at page 8, line 234.

**(A) Reviewer 2: Page 7, line 221.**

from where this value? / from where?

**(B) Response:** These values were described by Frank et al. (2015), however upon re-reading the paragraph above Equation 6, we realize that we should add more details in the next version of the manuscript. Additionally, we propose adding "Frank et al. (2015)" at the end of the sentence to make the origin more clear to the reader.

**(C) Changes:** We added the reference to our first entrainment model study "(Frank et al., 2015)" (where all details can be found) at the end of the two sentences to make the origin of the values and factors of the entrainment model more clear to the reader (page 8, line 232 and line 236). We also added a few words in this paragraph to give more details (page 8, line 232-236).

**(A) Reviewer 2: Page 7, lines 229-230.**

check numbering of eqs

**(B) Response:** Thank you for pointing out the error in the reference to the equation, we will correct and verify all equation numbers when preparing the next version of the manuscript.

**(C) Changes:** Error fixed at page 9, lines 243 and 244.

**(A) Reviewer 2: Page 9, line 281.** -2.5 ?

**(B) Response:** We agree with your suggestion we will also change the value to SI units, so -0.025 m/s, also for other occurrences of $\frac{dz}{dt}$ values in the manuscript.

**(C) Changes:** We changed the erosion rate values to SI units, so -0.025 m/s, for all other occurrences of $\frac{dz}{dt}$ values in the manuscript.

**(A) Reviewer 2: Page 10, line 314.**

how this was fixed?

**(B) Response:** The parameter $\xi$ was determined by varying it within the range proposed by the developers of the RAMMS model ($\xi = 100, 200, 400$) and inspecting the results. The only realistic velocities (in the steep ($\approx 60\%$) study reach of the Meretschibach channel) are obtained using $\xi = 200$ when combined with the variation of parameter $\mu$ ($= 0.5, 0.6, 0.7$). This is explained in the manuscript on page 10, lines 312-316. However to ensure that this is clear, we propose adding a sentence to clarify this procedure.

**(C) Changes:** We added a sentence (page 11, lines 331-333) to clarify the calibration and model set up procedure. We also updated the previous sentence in the manuscript (page 11, lines 327-329) to eliminate an inconsistency in how the model was actually applied in this case (e.g. it was correctly described in section 3.3.2. (page 10, lines 288-292) and in the caption of Fig. 6 on page 30, lines 794-797).

**(A) Reviewer 2: Page 13, line 432.**

Alternative, but related definition is that of Hungr, i.e. landslide growth rate = Vfinal / Vinitial

**(B) Response:** Thank you for pointing out Hungr's definition. We will verify which metric are used in the other papers which we reference, and for the next version of the manuscript we will choose the most suitable metric (as well as cite Hungr's landslide growth factor).

**(C) Changes:** We followed your suggestion for an alternative definition of the metrics and we now use Hungr's *volume growth* (VG) instead of the *bulking factor* (BF) (pages 13-14, lines 402-411

and new Figure 8). The VG values in the text were adjusted accordingly to Figure 9 (formerly

Figure 8). However this change has not changed the interpretation of the results or the discussion thereof (pages 16-17).

**Comments by the Editor (referencing comments from Reviewer 1 and 2)**

**(A) Editor: Reviewer 1, general comment:** additionally, provide also a short review on theoretical concepts explaining this behavior and how it is related to your model

**(B) Response and (C) Changes:** As mentioned above in the response to the general comments by reviewer 1, we edited sections 4.2. (calibration, on pages 11 and 12) and 5. (discussion, starting at page 14) for clarity . In section 5 we also added a discussion about the inherent feedback of in our entrainment model and its relation to the similar observations in laboratory and field studies (page 17, lines 535-541). We added several new comments and references to related discussions in our previous manuscript where the model was introduced (Frank et al., 2015). Recent entrainment modeling papers (e.g. Cuomo et al., 2016) are now referenced (page 15, line 463) and discussed in relation to our empirical approach (these references were suggested in some specific comments by reviewer 2).

**(A) Editor: Reviewer 1: Page 6, 195-196.:** I think just deleting the sentence is not a good solution. You have to explain your choice for the parameter of internal friction, It seems that you did not used the typical criteria for Voellmy friction

**(B) Response and (C) Changes:** We have not measured angles of internal friction and we cannot back-up this statement with observations, so we think that it is appropriate to remove this sentence from the manuscript because it may be confusing to readers. We edited the manuscript to clarify that in fact our calibration procedure for the selection of µ does not differ significantly from that proposed in the standard RAMMS model (e.g. the typical criteria for Voellmy friction). To make this clear, we added a citation to Bartelt et al. (2013) and a clear statement on the calibration of runout using the RAMMS model with entrainment (pages 14-15, lines 431-450).

**(A) Editor: Reviewer 1: Page 7, 221-222.:** please add the suggested figure

**(B) Response and (C) Changes:** We added the suggested figure showing the model sensitivity to critical shear stress. It's now presented as the new Figure 8. The former Figure 8 is now Figure 9. The (new) Figure 9 was also adapted to show the *volume growth* instead of a *bulking factor*, as suggested by Reviewer 2. Please also see the more specific comments on the changes made (shown above in "**(C) Changes**" at page 4, lines 103 ff. in this document).

**(A) Editor: Reviewer 2: Page 1, lines 23-24.:** provide a better reasoning for your calibration procedure and highlighting also the pro and cons of the chosen procedure

**(B) Response and (C) Changes:** We added some additional explanation in the text (also summarized in the abstract, page 1, lines 24-27) to clarify our calibration procedure, e.g. in section 4.1 (page 11, lines 331-333). Also, we discuss the benefits and limiting factors in the discussion (section 5, page 15, lines 442-450). In the text we also refer to an extensive discussion of exactly this topic (page 14, lines 440-441), which was extensively described by Frank et al., (2015).

[revised manuscript text omitted]

**A.** subcatchment "Bochtür", relevant part for debris flows lateral water supply, insignificant lakes, water supply, hydrologically disconnected debris slopes with active gravitational processes (2014)

main study area within subcatchment "Bochtür"

debris flow deposits, 07/20/2014

debris flow deposits of events on 07/28/2014 and 7/29/2014

geophones > debris flow timing, triggering of point radar point radar > central flow height meteorological stations intervall cameras lateral dams     sediment trap

**Figure 1. A.** Location of the Meretschibach catchment in Southern Switzerland. **B.** Subcatchments of the Meretschibach and locations of the instrumentation site and data available for the erosion model analyses **C.** Initiation zone of the July 2014 events and camera positions. The main study channel reach for the model testing is located in the middle part of "Bochtür" (black-white
retangle), swissimage©2014, swisstopo (5704 000 000) (2014).

[Figure]

**Figure 2. A.** Location of the Bondasca catchment in south-eastern Switzerland close to the border
to Italy. **B.** Perimeter of the 27 December 2011 rock avalanche deposit, including the main
deposition area (yellow polygon) and the deposits lower-elevation deposits which have been
partially exposed to erosion by debris flows in 2012 (red polygon). The 2012 post-event digital
elevation model (lidar, blue polygon) is from 18 July 2012 (data courtesy of the Amt für Wald,
Canton Graubünden). Pre-event digital elevation model (lidar) for 2009 is from the SwissAlti3D
(version 2012) data set from swisstopo, ©2012, swisstopo (5704 000 000) . The grey solid arrow
indicates the main debris-flow channel formed in 2012.

[Figure]

**Figure 3.** Calibration of modelled erosion patterns (**B1 to B3**) to the observed erosion depths (**A**) in the upper open debris slopes of the "Bochtür" catchment (Meretschibach) by varying values for the friction parameter μ. The blue polygon demarks the area where a differential DTM is available.

[Figure]

**Figure 4.** Erosion model configuration for the model simulations showing the initial block release
areas in the Meretschibach catchment (**A**) and the Bondasca catchment, Switzerland (**B**). The
hillslope is erodible within the brown shaded polygon.

[Figure]

**Figure 5.** Overview of rock avalanche deposits, subsequently formed debris flow channels, and the resulting overall elevation change in the Bondasca catchment (A, B). The elevation change map 2009 to 2012 (C) includes both the rock avalanche ( ≈ 1.5 Mio m³ on 27 Dec. 2011) and the first two  debris-flow events (5 and 14 July 2012).

[Figure]

**Figure 6.** Comparison of runout patterns at "Bochtür" in the Meretschti catchment. The debris flow
modeling is conducted using a (subtract) block release volume of (**A**) 10 m³ and no-entrainment
modeling, of (**B**) 10 m³ and entrainment modeling as well as a total (subtract) block release volume
of (**C**) 1,555 m³ (sum of release and eroded volume from (**B**)) and no-entrainment modeling.

[Figure]

**Figure 7.** Sensitivity of modeled erosion volume to initial block release volume in the
Meretschibach and in the Bondasca catchments.

[Figure]

**Figure 8.** Sensitivity of the volume growth VG = $(V_{ini} + V_{ero}) / V_{ini}$ to the critical shear stress $\tau_c$ depending on 5 different initial (block release) volumes $V_{ini}$ as set up based on two release areas in the Meretschibach catchment.

[Figure]

**Figure 9.** The volume growth VG = $V_{ini}$ +$V_{ero}$ / $V_{ini}$ consisting of the  sum of  the erosion volume $V_{ero}$ [m³]  and initial block release volume $V_{ini}$ [m³] per initial block release volume $V_{ini}$ [m³] and adressing three different erosion rates  for the Meretschibach and Bondasca catchments.